# Navigating the Cytokine Seas: Targeting Cytokine Signaling Pathways in Cancer Therapy

**DOI:** 10.3390/ijms25021009

**Published:** 2024-01-13

**Authors:** Noyko Stanilov, Tsvetelina Velikova, Spaska Stanilova

**Affiliations:** 1Medical Faculty, Trakia University, 6000 Stara Zagora, Bulgaria; noyko.stanilov@gmail.com; 2Medical Faculty, Sofia University St. Kliment Ohridski, 1 Kozyak Str., 1407 Sofia, Bulgaria; 3Department of Molecular Biology, Immunology and Medical Genetics, Medical Faculty, Trakia University, 6000 Stara Zagora, Bulgaria; spaska.stanilova@abv.bg

**Keywords:** cytokine signaling, cancer therapy, tumor microenvironment, immunotherapy, JAK/STAT signaling, TGF-β, NF–κB, epithelial mesenchymal transition, IL-6, inflammation, novel therapies, immune modulation

## Abstract

Cancer remains one of the leading causes of morbidity and mortality worldwide, necessitating continuous efforts to develop effective therapeutic strategies. Over the years, advancements in our understanding of the complex interplay between the immune system and cancer cells have led to the development of immunotherapies that revolutionize cancer treatment. Cytokines, as key regulators of the immune response, are involved in both the initiation and progression of cancer by affecting inflammation and manipulating multiple intracellular signaling pathways that regulate cell growth, proliferation, and migration. Cytokines, as key regulators of inflammation, have emerged as promising candidates for cancer therapy. This review article aims to provide an overview of the significance of cytokines in cancer development and therapy by highlighting the importance of targeting cytokine signaling pathways as a potential therapeutic approach.

## 1. Introduction

To escape immune surveillance, developing tumors use different mechanisms to dysregulate the antitumor T-cellular response, including the blockage of immune checkpoints such as programmed cell death protein 1 (PD-1), programmed cell death 1 ligand 1 (PD-L1), and cytotoxic T-lymphocyte-associated protein 4 (CTLA-4).

Cytokines are a group of small proteins with low molecular weight (<40 kDa) secreted by various immune cells, including Th-cells, dendritic cells, neutrophils, macrophages, B-lymphocytes, and others [1]. They serve as crucial mediators of communication and coordination between immune cells for regulating the development and type of the immune response, and maintaining immune homeostasis and surveillance. Cytokines regulate the differentiation, growth, proliferation, and function of various types of cells, and they are of fundamental importance for the realization of protective immune responses, as well as in pathological conditions dependent on inflammatory or autoimmune processes, infections, cancer, and trauma [1].

Cytokines are a diverse group of proteins that are divided, based on their receptors’ conformation, into transforming growth factors (TGFs), interleukins (IL), interferons (IFNs), colony-stimulating factors (CSFs), tumor necrosis factors (TNFs), and various chemokines [2]. When they bind to specific receptors in mono or dimeric form, they activate the intracellular receptor domains responsible for intracellular signaling by certain kinases. This intracellular signaling leads to the activation and nuclear translocation of distinct transcription factors that affect specific gene expression [3]. Many IL and IFN receptors act with Janus kinases (JAKs), subsequently involving transcription factors from the protein family of Signal Transducer and Activator of Transcription (STAT) [4]. The other signaling pathway starts when TNF binds to its cognate receptor, activating nuclear factor κB (NF–κB) and AP-1 [5].

Classifying cytokines considering their effect on inflammation divides them into proinflammatory and anti-inflammatory.

Chronic inflammation upregulates several proinflammatory cytokines, mainly IL-1, TNF-α, IFN-γ, IL-6, IL-12, IL-23, and IL-17, involved in both the initiation and progression of cancer [6,7,8,9]. Anti-inflammatory cytokines, mainly TGF-β (transforming growth factor- β) and IL-10, have a dual role in cancer initiation and progression. For this reason, the chronic inflammatory process is accepted as the seventh hallmark of cancer development [10].

A hypothesis established by Lippitz [11] a decade ago speculates that the cancer-specific and histology-independent uniform cytokine cascade is one of the manifestations of the underlying paraneoplastic systemic disease, linking the stage of cancer with both the functional status of the immune system and the patient’s prognosis. Neutralizing this cytokine pattern could offer novel and, as of yet, unexplored treatment approaches for cancer. Furthermore, although immunostimulatory cytokines are involved in local cancer-associated inflammation, cancer cells seem to be protected from immunological eradication by cytokine-mediated local immunosuppression and a resulting defect of the interleukin 12-interferon-γ-HLA-DR axis, where cytokines produced by tumors might have a pivotal role in this defect [11].

Generally, it has been accepted that the inflammatory microenvironment potentiates the transition of adenoma into carcinoma. The inappropriate regulation of the signaling pathway, as well as the altered expression of various proinflammatory transcription factors, mainly NF-κB, STAT families, and activating protein-1 (AP-1), have been reported as stimulators in tumor development and progression [12,13,14].

In addition, many intracellular signaling pathways, such as canonical and noncanonical Wnt, Notch, ErbB, Hedgehog, JAK/STAT (Janus kinase/signal transducers and activators of transcription), PI3K/Akt/mTOR (phosphoinositide 3-kinase/Akt/mammalian target of rapamycin), and TGF/SMAD, have been shown to play a pivotal role in tumorigenesis. 

Given their pivotal role in immune regulation and intracellular signaling, cytokines have emerged as critical players in cancer therapy. While cytokines hold great promise in cancer therapy, their pleiotropic nature and complex signaling networks pose challenges in achieving optimal therapeutic outcomes. Targeting cytokine signaling pathways provides a strategic approach to modulating cytokine-mediated effects and enhancing therapeutic efficacy.

This review article provides an overview of the significance of cytokines in cancer therapy, highlighting their multifaceted roles in modulating immune responses, tumor growth, and therapeutic interventions.

## 2. Cytokines Induced NFκB Signaling Pathway That Promotes Tumor Growth

Nowadays, we accept inflammation as a protective immune response against internal or external tissue-damaging agents designed to eliminate them and restore tissue homeostasis. That is the role of acute inflammation, whereas chronic (unresolved) inflammation is usually associated with tissue pathology. Chronic inflammation due to environmental signals is strongly associated with cancer risk [15]. Aggarwal et al., 2009, reported that approximately 20% of cancer could be due to chronic infection by various pathogens, 35% could be due to chronic infection by inhaled pollutants, including tobacco smoke, and 20% to obesity-related chronic inflammation [16]. Extracellular signals can promote inflammation by pathogen-associated molecular patterns (PAMPs) for environmental agents and damage-associated molecular patterns (DAMPS) receptors for endogenous agents [16]. 

In the initial stage of inflammation, the cells of the innate immunity secrete the cytokine triad of IL-1, TNF-α, and IL-6. Another early proinflammatory cytokine secreted mainly by antigen-presenting cells is IL-12. All these cytokines are upregulated at mRNA and protein levels in various tumors [17,18,19].

There are 11 members of the IL-1 family of cytokines and 10 members of the IL-1 family of receptors [20]. When IL-1 (IL-1α or IL-1β) binds to their cognate receptor, it forms a trimeric complex, including a co-receptor, which allows activation of intracellular Toll-IL-1-Receptor (TIR) domains and recruiting MyD88. This triggers a kinase cascade and subsequent activation of NFκB. The TIR domains of the IL-1R family are the same as the TIR domains in each TLR, responsible for inflammation induced by various pathogens. All members of the IL-1 family exhibit the same mechanism of intracellular signal transduction [21].

The NFκB family of transcription factors includes subunits—cRel, RelA (p65), RelB, p50, and p52, which can form different heterodimers with p105 (p105/p50—NFκB1) and p100 (p100/p52)—NFκB2 [22]. In the cytoplasm of non-stimulated resting cells, the inactive form of NFκB exists as a heterodimer that binds with one of the inhibitory IκBα, IκBβ, and IκBε subunits. 

During intracellular signal activation by MyD88, IκB undergoes phosphorylation by the IκB-kinase complex, followed by degradation, releasing the active form of the most common NFκB dimer consisting of p50 and p65 subunits, also known as canonical NFκB. Noncanonical NFκB is the dimer p52/relB transcription factor, and it is activated mainly by TNF-receptor (TNFR) family members [23].

The active transcription factor, both canonical and noncanonical NFκB, translocates to the nucleus, where it binds to the specific promoter sequence of target genes, and turns transcription of the responding genes, thus producing certain cellular responses. Some of the responding genes encode other inflammatory cytokines, such as TNF- α and IL-6, and other genes control apoptosis (BCLXL, BCL2, BCLXS, XIAP) and angiogenesis (VEGF) [24]. 

Sustained activation of the NFκB signaling pathway plays an essential role in inflammatory-associated cancer [25]. NF-κB is constitutively active in most tumor cell lines, and it is required for the proliferation of these cells. Constitutive NF-κB activation has also been reported in patient samples of many solid and blood tumors. For example, in breast cancer, it has been reported that NF-κB activation is at approximately 90% in grade III compared to 38% in first-grade tumors [26]. Moreover, a variety of malignancies have been reported in association with mutated genes encoding subunits of NFκB or IκB proteins [27].

The typical molecular inflammatory signature in the tumor microenvironment includes both proinflammatory cytokine (IL-1, TNF-α, IL-6, and IL-8) and chemokines (CCL20, CXCL13, and CXCL8). The increased secretion of proinflammatory proteins realizes a positive feedback loop to accelerate the activation of the NFκB pathway, leading to the upregulation of tumor growth. It is also important to note that the NF-κB signaling pathway intersects with many other signaling pathways relevant to tumorigenesis, such as AP1, STAT3, Notch, WNT–β-catenin, and p53.

Therefore, the NFκB signaling pathway contributes to tumor cells’ survival and proliferation, which turns this pathway into a target for antitumor therapy [28].

Figure 1 illustrates the intricate web of cytokine signaling pathways, showcasing their pivotal roles in both inflammation and tumor development, providing a comprehensive visual overview of the dynamic interplay between the immune system and tumorigenesis.

Recently, a lot of natural and synthetic NFκB inhibitors have been tested for anticancer activity in different experimental cancer models [29,30,31], but none have applied for treatment. Evidently, this approach still requires a lot of improvements and further research to validate and optimize antitumor drugs.

The central role of proinflammatory cytokines and their receptors in NFκB activation and tumor development also makes them potential therapeutic candidates for cancer treatment. One possibility is using monoclonal antibodies against proinflammatory cytokines or their receptors. Recombinant IL-1Ra produced from E. coli (generic anakinra) is fully active in blocking the activities of IL-1α and IL-1β. It is used mainly in the treatment of rheumatoid arthritis and some other autoimmune diseases [32]. Lust et al. (2013) [33] used this IL-1 receptor antagonist for the first time for the treatment of early-stage multiple myeloma. Ron and Voronov (2017) [34] discuss the role of IL-1 agonistic molecules in tumor progression and their potential to serve as targets in antitumor immunotherapeutic approaches.

## 3. Cytokine-Induced STAT Signaling Pathway Promoting Tumor Invasion and Metastasis

The increased and sustained expression of many cytokines can cause the activation of the evolutionarily conserved transmembrane signal transduction of the Janus kinase/signal transducer and activator of transcription (JAK/STAT) signaling pathway. This ligand-receptor complex activates one of four tyrosine protein kinases: JAK1, JAK2, JAK3, and TYK2, acting in the cytoplasm. JAKs are widely expressed and noncovalently bind to cytokine receptors (CRs) having the γc subunit or CRs with gp130 subunits, or R for IFN-α/β [35,36]. They mediate tyrosine phosphorylation of the intracellular domain of these Rs, which recruits one or more corresponding proteins from the STAT family. In humans, the STAT family includes seven members of transcription factors: STAT1, STAT2, STAT3, STAT4, STAT5A, STAT5B, and STAT6, encoded by genes located on chromosomes 2, 12, and 17 [36,37,38]. Tyrosine-phosphorylated STATs form homo- or heterodimers that enter the nucleus, where they bind with specific promoter sequences to regulate the transcription of various target genes. Dysregulated JAK-STAT signaling has been identified in diverse immune-mediated conditions, including cancer development and progression.

Among cytokines, the main activator of the JAK/STAT signaling pathway is IL-6, which activates STAT3 after binding with ubiquitously expressed GP130, a β-subunit of the IL-6 receptor. All other cytokines of the IL-6 family and some other cytokines and growth factors (mainly EGF family members) through their receptors can activate STAT3. The latter is the most prominent and widely studied STAT transcription factor, because its mutation and overactivation play a central role in cancer cell proliferation, metastasis, and immune evasion [39,40,41]. 

STAT-3 activates *c-myc* protooncogene and induces the enhanced expression of cyclin D1, cyclin B, and cdc2, which are involved in cell cycle regulation. Binding of cyclin D1 to cdk4 or cdk6 drives progression from G1 to S phase in epithelial cells and can cause uncontrolled cell proliferation [42,43].

The active form of phosphorylated STAT3 has been found to induce upregulated expression of cdc2, cyclin B1, m-ras, and E2F-1 in colon and breast carcinomas [44]. Simultaneously with the upregulation of cell proliferation, STAT3 activity enhances the expression of genes encoding antiapoptotic proteins such as Bcl-xL, McL-1, BcL-2, and survivin [43,45,46]. Thus, constitutive activation of STAT3 in cancer cells promotes cell survival and proliferation and inhibits apoptosis, which benefits tumor growth. 

The classical IL-6-JAK/STAT3 pathway is involved in inducing resistance to chemotherapy and therapy with aromatase inhibitors. The activity of this pathway positively correlates with poor survival in Hormone Receptor-positive (HR+) breast cancer [47,48]. Enhanced expression of IL-6 and persistent activation of STAT3 are strongly associated with chemotherapy sensitivity of high-grade or triple-negative breast cancers [41].

Moreover, active STAT3 can promote tumor metastasis in a variety of human cancers via the upregulation of VEGF and other angiogenic factors such as angiopoietin, chemokines mediating tumor invasion, and matrix metallopeptidase-9 [49,50]. New evidence reported by Sun et al. suggests the involvement of STAT3 activity in driving the epithelial-mesenchymal transition (EMT), which promotes extrahepatic cholangiocarcinoma metastasis [51].

The importance of the JAK/STAT pathway in tumorigenicity increases, bearing in mind the cross-talk between this pathway and others, such as NF-κB; Notch signaling, PI3K, MAPK signaling, and TGF- β signaling pathways, which are also involved in tumor development and metastasis [36].

The prominent role of JAK/STAT signaling, particularly STAT3 activity, in cancer development and progression across many types of tumors makes their activity a desired target for anticancer drug development. Targeting the JAK/STAT pathway can be performed by four groups of drugs, depending on their effect on the components of the signal transduction process: antibodies against cytokines, antibodies against receptors, inhibitors of JAK activity, and inhibitors of STAT activity.

Manipulation of upstream components, such as cytokines or their receptors, is performed with antibodies against cytokines, their cognate receptors, or engineered cytokines.

IL-2 has been approved for the treatment of metastatic renal cell carcinoma and metastatic melanoma. Engineered cytokines, such as TNF-α-derived peptide and recombinant fusion IL-2 (ALKS 4320), have demonstrated less toxicity and effective inhibition of tumor growth in preclinical studies [52,53]. A new class of biopharmaceuticals that can potentially improve cytokine’s therapeutic index are antibody-cytokine fusion proteins, having been tested in preclinical cancer models and clinical trials [54].

Given the significant role of IL-6 in the constitutive activation of STAT3, some therapeutic approaches to downregulate the IL-6 signaling pathway have become targets for the therapies of various types of cancer. Among them, monoclonal antibodies against IL-6, such as Siltuximab, and monoclonal antibodies against IL-6R, such as Tocilizumab, alone or in combination with conservative chemotherapy, appear as potential therapeutic approaches for effective anticancer therapy [55]. In HER-positive breast cancer patients, affecting the IL-6/JAK2/STAT3 axis by trastuzumab therapy alone or in combination with ruxolitinib, a selective JAK1/2 inhibitor, significantly improves clinical outcomes, as reported by Rodriguez-Barrueco et al. [56].

Most JAK inhibitors are designed to have ATP competitive activity; thus, they are non-selective, inhibiting multiple JAK isoforms and displaying many adverse events. They are more suitable and valuable for treating inflammatory and some autoimmune diseases than cancer. A comprehensive review summarizes the current knowledge of JAK inhibitors and their clinical applications and adverse effects [57]. 

Chikuma et al. also demonstrated that the suppressors of the cytokine signaling (SOCS) family of proteins (including CIS, SOCS1, SOCS3), which are major negative regulators of the JAK/STAT pathway, are promising third immune checkpoint molecules because they control CD4+ T cell polarization and CD8+ T cell maturation [58]. Moreover, applying shRNA and SOCS anti-sense oligonucleotide for SOCS1 inhibition has revealed therapeutic antitumor effects [58]. 

The most suitable target for decreasing cytokine-induced cancer cell proliferation is STAT3 activation. Moreover, STAT-3 activation may contribute to the development of drug resistance, particularly in chemotherapies aimed at initiating apoptosis. To prevent tyrosine phosphorylation of STAT3 and reduce its transcriptional activity, a variety of compounds with such kinds of inhibitory potency has been identified. 

Over the last decade, different strategies, including small molecules either isolated from natural sources or synthetically produced, anti-sense oligonucleotides, and gene therapy, have been utilized to downregulate STAT3 signaling and thereby inhibit cancer cell proliferation and invasion [39,59,60].

## 4. TGF-β Signaling Pathway

Transforming growth factor-beta (TGF-β) is a ubiquitously expressed cytokine with many regulatory functions in maintaining tissue homeostasis in vertebrates, discovered by Todaro et al. [61]. As indicated by the name of this cytokine, one of its first-known roles was to “transform” normal cultured fibroblasts into abnormal, progressively growing colonies, thus marking its role in carcinogenesis. Much recent evidence has documented both tumor-supportive and suppressive roles of TGF-β signaling pathways in human cancer development. 

The TGF-β family includes TGF-β1, TGF-β2, and TGF-β3, activins (Act), inhibins, bone morphogenetic proteins (BMPs), and growth differentiation factors—GDFs, all of which are dimers [62].

TGF-β1 is mostly found in human tissues in its active form, which, after post-translational processing, is a homodimer. It consists of two polypeptide chains, each containing 112 amino acid residues connected by a disulfide bond, forming a complex with a total molecular weight of 25 kDa [63].

Beyond the role of TGF-β as an immunoregulatory and anti-inflammatory cytokine, it is involved in controlling cell proliferation, differentiation, apoptosis, cell migration, and epithelial-mesenchymal transition (EMT) [64,65].

After the release of TGF-β1 from the extracellular matrix, it interacts with a heterotetrameric receptor complex consisting of two TGF-β receptor I (TβRI, TGFBR1) and TGF-β receptor II (TβRII, TGFBR2) subunits each. The TGFBR1 and TGFBR2 receptor subunits are transmembrane glycoproteins that contain an N-extracellular terminus with a ligand-binding domain, a transmembrane portion, and an intracellular C-terminus containing a domain with serine/threonine kinase activity [66,67].

The activated TGF-β ligand binds to TGFBR2 on the cell surface, resulting in binding to TGF-β receptor I subunits, where the serine/threonine kinase activity domain of TGFBR2 phosphorylates the GS domain of TGFBR1. A heterotetrameric ligand-receptor transmembrane TGF-β complex is formed with the type I receptor, which induces a SMAD-dependent (canonical signaling) or SMAD-independent (noncanonical signaling) intracellular signaling pathway. This complex can be stabilized by TGF-β co-receptors, such as beta glycan (TGF-β type III receptor, TβRIII) or endoglin (CD105), which bind TGF-β ligands to enhance signaling [68].

SMAD proteins are well-conserved transcription factors that can regulate target genes positively or negatively. This family includes R-Smads, receptor-regulated proteins, a common mediator named Co-Smad (Smad-4), and inhibitory Smad-6 and 7. Both R- and Co-Smads have two conserved Mad homology domains at the ends of their polypeptide chain, namely MH-1 and MH-2, respectively [69]. Inhibitory Smads also have a C-terminal MH2 domain but lack the N-terminal MH1 domain. They act as negative feedback regulators of TGF-β signaling by preventing the formation of R- and Co-Smad active complexes (Smad-6) or by promoting ubiquitin-mediated degradation of the active TGFBR complex (Smad-7) [70].

The activated TGFBR1 receptor phosphorylates SMAD2 and SMAD3, forming a complex SMAD2/SMAD3 and recruiting Smad-4 through their MH-2 domains. Because of the nuclear localization sequence in the MH-1 domain, the formed complexes translocate to the nucleus. The MH-1 domain is also responsible for binding with co-activators or other transcriptional nuclear factors and interaction with DNA [70,71,72]. In the nucleus, Smad complexes are transactivated with the help of MH-2 domains to regulate the transcription of target genes, encoding proteins implicated in cell proliferation, apoptosis, and differentiation.

Inhibitory SMADs antagonize TGF-β signaling by inhibiting R-Smad binding sites (GS domains) on TGFBR1 in the absence of the TGF-β ligand [62,73].

Depending on the proteins attracted to the ligand-receptor complex, binding of TGF-β to its receptors can activate other SMAD-independent signaling pathways (noncanonical signaling) by stimulating various kinases such as JNK, P38, ERK, MAPKs, Rho/ ROCK, or phosphatidylinositol-3 kinase (PI3K), regulating many cellular functions. These kinases are capable of phosphorylating the linker regions of SMAD2/3, which is the region between the N-terminal MH1 domain and the C-terminal MH2 domain of the SMAD proteins [74].

The tumor suppressive activity of TGF-β is manifested at the early stage of tumorigenesis generally by arresting cell proliferation and promoting apoptosis in malignant cells. In normal and premalignant epithelial cells, TGF-β signaling leads to the activation of CDK inhibitors p21 and p15, which can stop the cell cycle in the G1 phase and thus, suppress the proliferation of tumor cells [73,75,76].

The tumor-suppressive role of TGF-β signaling via the canonical pathway is mediated by Smad2 and Smad3. Genes for both Smad2 and Smad3 can be accepted as tumor suppressor genes [77,78]. Smad-4 is a critical player in the EMT transition required for metastasis in cancer progression, particularly for bone metastasis of breast cancer [65]. TGF-β signaling can also downregulate antiapoptotic proteins from the Bcl-2 family and subsequently induce apoptosis. The tumor-suppressive role of TGF-β via the noncanonical pathway is promoted by activating caspase-8, which can trigger apoptosis [65]. 

In addition to its antitumor properties, it should be noted that TGF-β is a potent anti-inflammatory and regulatory cytokine that plays a prominent role in immune tolerance, and it can suppress tumor-elicited inflammation. Carcinomas that express high levels of TGF-β may escape immune surveillance. TGF-β1 is a regulatory cytokine that orchestrates the differentiation of both T-regulatory (Tregs) and T-helper 17 (Th17) cells in a concentration-dependent manner [79]. Low doses of TGF-β1 induce proinflammatory Th17 cell differentiation. In contrast, high doses are able to maintain pro-survival through up-regulation of Foxp3 expression in peripheral CD4 + CD25+ regulatory T cells [80,81]. 

TGF-β pathway dysregulation in advanced cancer stages leads to cell reprogramming, by which cancer cells enhance survival and metastasis in other tissues, which turns TGF-β into a tumor promoter. The main protumor activity of the TGF-β pathway is mediated by stimulating EMT, angiogenesis, and immune escape [64,82,83,84]. 

In addition, TGF-β modulates the antitumor response towards the differentiation of preferential Treg cells instead of the Th1 antitumor subpopulation [85,86].

Like other intracellular signaling pathways involved in tumor growth, the TGF-β pathway’s different components are promising targets for cancer therapy. Neutralizing monoclonal IgG antibodies against both TGF-β (fresolimumab) and TGFBRII (LY3022859) have demonstrated a significant decrease in tumor growth and metastasis in animal models and phase-II clinical trials [87,88].

In addition to neutralizing antibodies, some TGF-β ligand capture molecules, such as AVID200, have been designed and tested in phase I clinical trials [89]. Small molecule inhibitors and potent ATP-competitive TGF-βRI inhibitors are another strategy to target this pathway. A new strategy to specifically block gene expression driven by TGF-β signaling is using antisense oligonucleotides. Some of these antisense oligonucleotides (e.g., trabedersen) are designed to block the transcription of TGF-βRII mRNA and have been tested in patients with colorectal cancer, pancreatic cancer, and melanoma [90].

Recent studies have provided evidence that TGF-β signaling is also involved in metabolic reprogramming in cancer cells, favoring tumor growth, and can be targeted [91].

The TGF-β signaling pathway intersects many other signaling pathways, such as NF-kB, PI3K/Akt, and IL-6/JAK2/STAT3 signaling pathways [83,92,93]. This cross-talk between cytokine signaling pathways orchestrates cellular function in a finely balanced manner, and dysregulation associated with malignant transformation initiated in one signaling pathway interferes throughout the network. We must keep in mind that targeting one signaling pathway may affect at least two or three others, which may result in reduced or adverse effects. The TGF-β signaling is an excellent example of the context-dependent manner of regulation with diverse biological effects on cell behavior. Obviously, we need a deeper understanding of cytokine signaling networks and their role in cancerous cell transformation.

## 5. Cytokine-Induced Signaling Pathway Related to Epithelial-Mesenchymal Transition and Tumor Metastasis

The EMT is a process tightly related to cancer development based on ongoing chronic inflammation. Furthermore, it has been shown that cytokines and chemokines, as we already mentioned, that is, TGF-β, can mediate EMT changes. leading to cancer cell development [94,95]. 

A systematic review by Ray et al. (2023) demonstrated the role of various cytokines, such as TGF-β, TNF-α, IL-6, IL-1β, and IL-8, in promoting EMT in ovarian, endometrial, and cervical cancers. The authors also emphasized the novel therapies targeting EMT signaling pathways to prevent the progression, metastasizing, recurrence, and drug resistance of these gynecological cancers [96]. 

Interestingly, IL-9, produced by T-helper 9 cells, was shown to downregulate N-cadherin and vimentin expression in cervical cancer while upregulating E-cadherin expression. These antitumor properties of IL-9 are promising in preventing metastasis and cervical cancer progression [97]. 

Additionally, Zhang et al. demonstrated EMT induced by CCL20 through the Erk1/2-Akt pathway [98]. They focused on astrocyte-elevated gene-1 (AEG) expression, an oncoprotein with multiple actions, that could be targeted in cervical cancer.

Regarding endometrial cancer, EMT has been associated with TGF-β, IL-6, CCL18, RANK/RANKL/CCL20, CXCR4/CXCL12, etc. [95], where new treatment possibilities target TGF-β (i.e., a flavonoid Isoliqueritigenin, fluorene-9-bisphenol, etc.) [99,100].

EMT signaling pathways have also been investigated in breast cancer metastasis and invasion, including TGF-β, TNF-α, Notch, Wnt, Hedgehog, and receptor tyrosine kinases, converging on common transcriptional factors (i.e., Snail, Slug, Twist, and ZEB1/2). The authors concluded that understanding EMT signaling pathways may help establish better treatment options for breast cancer patients [101].

Cytokine involvement in EMT and potential treatment strategies were also shown in bladder cancers, and the treatment options included IL-6, IL-8, and TGF-β treatments [102].

## 6. Targeting Cytokine Signaling in Cancer Therapy: Objectives

The number of clinical trials investigating the safety and effectiveness of cytokine-based medications, both alone and in combination with other immunomodulatory therapies, has skyrocketed due to revived interest in the antitumor effects of some cytokines [103,104].

Starting as monotherapy with IL-2 or IFN-α for several malignant diseases (i.e., advanced renal cell carcinoma, metastatic melanoma, AIDS-related Kaposi’s sarcoma, etc.), cancer immunotherapy has evolved into safer and more efficacious targeted therapy, immune checkpoint inhibitors, and anti-CD19 chimeric antigen receptor (CAR) T cell therapies [103]. It is worth mentioning that CAR T cells depend on cytokines for both in vitro (i.e., expansion) and in vivo viability (i.e., the persistence of transferred T cells) [105]. 

Berraondo et al. [103] stated three concepts for searching for new-generation cytokine-based cancer drugs: (1) using synergic combinations (i.e., anti-PD-1–PD-L1 monoclonal antibodies and CAR19 T cells combined with cytokines); (2) improving pharmacokinetic profiles to increase cytokine concentration in the tumor microenvironment (i.e., via conjugation with polyethylene glycol (PEG), constructing a fusion protein with antibodies, Fc domains, apolipoprotein A-I, albumin, or the latent peptide of TGF-β); (3) local administration of cytokines to increase their concentration in the tumor environment [103].

Rallis et al. complemented these strategies with (4) revealing context-dependent interactions in the tumor microenvironment to enhance cytokine effects; and (5) elucidating the role of cytokine genetic polymorphisms [106]. 

Eventually, cancer immunotherapy based on cytokines relies on potentiating the immune response toward tumors by acting on each phase of the cancer immunity cycle and development. Cytokines can enhance antigen priming, increase the recruitment of cytotoxic T cells in the cancer, and improve their activities [107].

Heimberger et al. (2023) addressed the pan-cancer heterogeneity of cytokines. The authors emphasized that cytokines can mediate proinflammatory or immunosuppressive effects, in addition to having tumor-cytotoxic and tumor-promoting functions [108].

In line with this, gene therapy techniques that involve the use of adenoviruses as transfer vectors expressing TNF-α, IL-2, IL-12, IFN-a2b, IFN-β, or GM-CSF have been used to modulate cytokine expression in several cancer indications [108].

A novel approach in cancer immunotherapy is targeting cytokines signals to enhance γδT cells. This subpopulation can be polarized in the tumor environment under cytokine influence, thus contributing to adaptive immune responses against tumor cells. Studies have shown that γδT cell function could be mediated, i.e., IL-2 plus IL-15, IL-12 plus IL-18, IL-27, IL-21 plus IL-2 contribute to enhanced cytotoxicity, TGF-β increases antitumor cytotoxicity in the presence of IL-2, etc. [109].

However, γδT cells’ role during tumor development is still unclear. There are three main challenges in targeting cytokine signals to improve γδT cell-based immunotherapy: (1) despite the good safety profile, some patients do not benefit clinically; (2) rapid exhaustion and decreased survival of γδT cells; (3) immunosuppressive mechanisms in tumors that impair γδT cells [109].

## 7. Recent Advancements in Cytokine-Based Cancer Treatments: Clinical Trials Data

Since cytokines can affect the innate and adaptive immune systems employed in cancer immunosurveillance and control, cytokines are an intriguing therapeutic tool. However, some of the drawbacks of cytokine-based therapies, such as pleiotropy and toxicities, can be overcome by chemically modifying them with polymers or peptides to increase tissue/cell specificity and cytokine half-life. Moreover, cytokines can also be engineered to reduce off-target effects and improve cytokine specificity, enhancing their efficacy and safety profiles [110].

A systematic review of Fu et al. (2023) revealed the recent progress in cytokine engineering strategies and preclinical/clinical therapeutics for cancer. It was demonstrated that new engineering strategies have been developed to increase tumor targeting, lengthen the therapeutic window, extend pharmacokinetic effects, and decrease side effects to improve treatment efficacy [111].

Atallah-Yunes et al. (2022) gathered data on the biological mechanisms of cytokine-based immunotherapy for cancer and lymphoma, paying attention to current clinical trials, challenges, and future perspectives [112].

Soler et al. also reviewed the up-to-date data on preclinical and clinical trials targeting the IL-6 cytokine family in cancer, including IL-6, oncostatin M, leukemia inhibitory factor, IL-11, IL-27, IL-31, ciliary neurotrophic factor, cardiotrophin 1, and cardiotrophin-like cytokine factor 1 [113].

Furthermore, it was shown that IL-6 cytokine signaling is associated with resistance to immunotherapy and poor prognosis in patients with multiple myeloma, melanoma, breast, ovarian, cervical, prostate, pancreatic, and lung cancer [114].

Recently, the need for predictive biomarkers to assess cytokine-based immunotherapy for cancer has led to robust investigations designed to find rational therapeutic strategies and suitable targets. This is especially valid since cytokines typically function in loops and collaborate with other cytokines and chemokines, and targeting many cytokines at once may occasionally provide more effective antitumor effects than utilizing cytokine-targeting medications in monotherapy [113].

To provide a comprehensive overview of the ongoing efforts to target cytokine signaling pathways in cancer therapy, Table 1 summarizes key clinical trials exploring novel interventions and therapeutic strategies related to cytokine-based immunotherapy in cancer.

## 8. Future Directions and Challenges in Emerging Cytokine Targets and Novel Therapies

Identifying novel cytokine targets holds immense promise as we navigate the frontier of cytokine-based cancer treatments. Ongoing research explores the nuanced roles of specific cytokines in tumor microenvironments, uncovering potential therapeutic targets. From manipulating antitumor responses to modulating immune-suppressive signals, these emerging targets pave the way for innovative therapies that can potentially revolutionize cancer treatment. In line with this, the era of personalized medicine has extended its reach to cytokine-based cancer therapies. Advancements in understanding individual variations in immune responses and cytokine interactions allow for tailoring treatments based on a patient’s unique profile. Personalized approaches aim to optimize therapeutic outcomes while minimizing adverse effects, marking a paradigm shift toward precision medicine in cytokine-based cancer treatment.

However, despite initial successes, resistance to cytokine-based therapies remains a formidable challenge. Unraveling the intricacies of resistance mechanisms, whether inherent or acquired, is paramount. Innovative strategies, including combination therapies and immunomodulatory interventions, are under exploration to overcome resistance hurdles. Addressing these challenges is essential for maximizing the efficacy and durability of cytokine-based cancer treatments, paving the way for more resilient and effective immunotherapeutic approaches.

## 9. Conclusions

In this review, we comprehensively explored the multifaceted impact of cytokine signaling pathways on cancer initiation, development, progression, and metastasis, highlighting key insights into their intricate roles within the tumor microenvironment. We conclude that promising strides in cytokine-based cancer therapies have become evident. Targeting these signaling pathways holds immense potential for reshaping cancer treatment landscapes, offering novel avenues for therapeutic interventions. However, ongoing research is needed. Prospects hinge on unraveling complexities, refining personalized approaches, and devising strategies to overcome challenges associated with cytokine-based immunotherapy. Continued research promises to unlock new frontiers in cytokine-based cancer therapies, fostering hope for enhanced treatment modalities and improved patient outcomes.

## Figures and Tables

**Figure 1 ijms-25-01009-f001:**
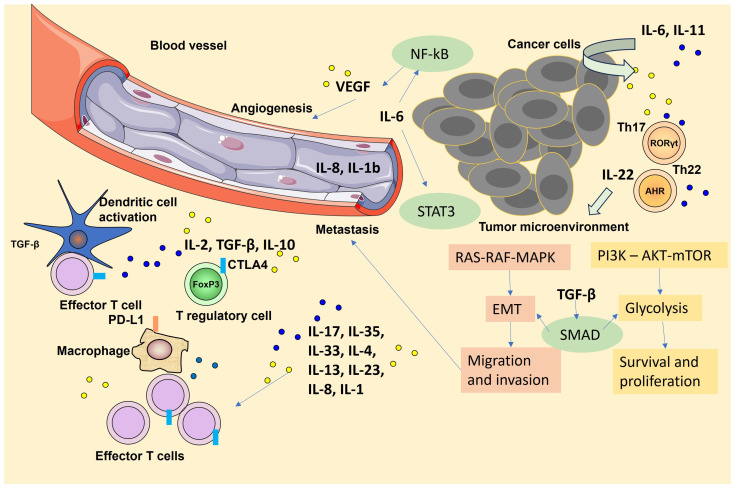
Cytokine signaling pathways and their pivotal roles in both inflammation and tumor development. Activation of nuclear factor-κB (NF-κB) via many stimuli (i.e., persistent inflammation in response to pathogen-associated molecular patterns (PAMPs) and danger-associated molecular patterns) leads to the production of proinflammatory cytokines and mediators by fibroblasts, epithelial cells, and myeloid cells (i.e., dendritic cells (DCs), monocytes, and macrophages). Additionally, tumor-initiating cells recruit macrophages, which, upon activator protein 1 (AP-1) signaling, produce transforming growth factor-β (TGF-β) that suppresses the function of cytotoxic T lymphocytes (CTLs). In response to IL-1β, other cytokines are also produced: IL-6, IL-11, and IL-22 from epithelial and myeloid cells. Furthermore, IL-6 activation of STAT3 signaling is observed in multiple cancer types and induces proliferation, survival, stemness, epithelial–mesenchymal transition (EMT), and migration of transformed cells. T helper 17 (TH17) cells differentiate under the influence of IL-1β, IL-6, and a higher concentration of TGF-β. These cells then release IL-17A and IL-17F (IL-17A/F) in response to IL-23 stimulation from DCs. IL-17 normally stimulates NF-κB to facilitate wound-healing signaling but can also aggravate nascent tumor development. Some cytokines, such as IL-6, TGF-β, or IL-11, may act in an autocrine manner to activate consequently signal transducer and activator of transcription 3 (STAT3), NF-κB, rat sarcoma (RAS)–rapidly accelerated fibrosarcoma (RAF)–mitogen-activated protein kinase (MAPK), and phosphoinositide 3-kinase (PI3K)–AKT–mTOR signaling to upregulate glycolysis and induce metabolic reprogramming. These pathways can lead to increased migration and epithelial–mesenchymal transition (EMT), increased proliferation, decreased apoptosis, increased migration, and the production of cytokines, including metalloproteinases, vascular endothelial growth factor (VEGF), and IL-8, which in turn induces angiogenesis. Parts of the figure were drawn using pictures from Servier Medical Art. Servier Medical Art by Servier is licensed under a Creative Commons Attribution 3.0 Unported License (https://creativecommons.org/licenses/by/3.0/). Copyrights: the authors.

**Table 1 ijms-25-01009-t001:** Clinical trials on cytokine-based interventions in cancer therapy.

Target Cytokine/ Cytokine Receptor	Intervention, Combinations	Cancer Type	Mechanisms of Immunotherapy	Clinical Trials
TNF-α	Nivolumab + ipilimumab + certolizumab or infliximab	Advanced melanoma	Blockade of activation-immune cell death of tumor-infiltrating lymphocytes	NCT03293784
TGF-β	Galunisertib + nivolumab; Galunisertib + durvalumab; Fresolimumab + radiotherapy; M7824 (dual block of TGF-β and PD-L1)	Advanced Hepatocellular Carcinoma, Metastatic Pancreatic Cancer, Early Stage Non-small Cell Lung Cancer, Previously Treated Advanced Adenocarcinoma of the Pancreas	Inflammation of immune-excluded tumours	NCT02423343 NCT02734160 NCT02581787 NCT03451773
LIF	AZD0171, durvalumab, gemcitabine, nab-paclitaxel; AZD0171, durvalumab, oleclumab, monalizumab, MEDI5752, dato-DXd, pemetrexed, carboplatin, cisplatin, paclitaxel	Locally advanced or metastatic solid tumors NSCLC	Blocking LIF/LIFR to inhibit tumor growth and DNA damage responses	NCT04999969 NCT05061550
CSF-1	Cabiralizumab + nivolumab; Pexidartinib + durvalumab; Pexidartinib + durvalumab or tremelimumab; Pexidartinib + pembrolizumab	Non-small Cell Lung Cancer or Renal Cell Carcinoma, Metastatic/Advanced Pancreatic or Colorectal Cancers, Advanced Solid Tumors, Advanced Melanoma and Other Solid Tumors	Suppression of tumor-associated myeloid cell	NCT03502330 NCT02777710 NCT02718911 NCT02452424
IL-2	Nivolumab; pembrolzumab, ipilimumab, enoblituzumab; TASO-001; NKTR-214 + atezolizumab; NKTR-214 + nivolumab; NKTR-214 + nivolumab + ipilimumab; Cergutuzumab amunaleukin + atezolizumab; RO6874281 + trastuzumab or cetuximab; RO6874281 + atezolizumab; RO6874281 + atezolizumab + bevacizumab	Melanoma, RCC, ovarian cancer, solid tumors	Expansion of NK and T lymphocytes	NCT03991130 NCT02964078 NCTO4562129 NCT04630769 NCT04862767 NCT03138889 NCT02983045 NCT03282344 NCT03435640 NCT02350673 NCT02627274 NCT03386721 NCT03063762
IL-6R	ERY974, tocilizumab, atezolizumab, bevacizumab, Ipilimumab, nivolumab, tiragolumab, conventional surgery, radiation, TPST-1120, RO7247669, Sarilumab, relatlimab, carboplatin, gemcitabine, linagliptin, ipatasertib, sacituzumab govitecan, evolocumab, etrumadenant, enfortumab vedotin, niraparib, Hu5F9-G4, cisplatin, gemcitabine, Obinutuzumab, cibisatamab, Capecitabine, SGN-LIV1A, gemcitabine + carboplatin or eribulin, selicrelumab, nab-Paclitaxel, cobimetinib, RO6958688, docetaxel, CPI-444, pemetrexed	HCC; Advanced Melanoma; NSCLC; Urothelial Carcinoma, Bladder Cancer; Recurrent Glioblastoma, Diffuse Astrocytoma; Advanced Liver Cancers; Pancreatic Cancer; Colorectal Cancer; Prostate Cancer; Head and Neck Squamous Cell Carcinoma; Triple Negative Breast Cancer.	Blocking IL-6 receptor to suppress cancer invasiveness and metastasizing	NCT05022927 NCT04940299 NCT04729959 NCT04691817 NCT04524871 NCT04258150 NCT03999749 NCT03869190 NCT03866239 NCT03821246 NCT03708224 NCT03424005 NCT03337698 NCT05428007
IL-6	Siltuximab, spartalizumab	Metastatic pancreatic adenocarcinoma	Blocking IL-6 receptor to suppress cancer invasiveness and metastasizing	NCT04191421
IL-8	mAb anti-IL-8 + nivolumab	Advanced Cancers	Suppression of tumor-associated myeloid cell	NCT03400332
IL-10	Pegilodecakin + FOLFOX	Metastatic Pancreatic Cancer	Blockade of activation-immune cell death of tumor-infiltrating lymphocytes	NCT02923921
IL-12	DC/tumor vaccine; cetuximab; pembrolizumab; Transduced TILs; Electroporated plasmid; Electroporated plasmid + pembrolizumab	Glioma, head and neck cancer, solid tumors	Promotion of NK cells and Th1 CD4+ and CD8+ lymphocytes	NCT04388033 NCT01468896 NCT03030378 NCT01236573 NCT01579318 NCT00323206 NCT01502293 NCT02345330 NCT02493361 NCT03132675
IL-15	Mogamulizumab, nivolumab, ipilimumab, avelumab; ALT-803; IL-15 + T or NK cells; IL15 + alemtuzumab; IL15 + rituximab	CTCL, solid tumors, PTCL	Expansion of NK and T lymphocytes	NCT04185220 NCT03388632 NCT03905135 NCT02989844 NCT01875601 NCT02465957 NCT01385423 NCT1369888 NCT02689453 NCT02384954
IL-27	SRF388, pembrolizumab SRF388, atezolizumab, bevacizumab, placebo	Advanced ccRCC or HCC, or anti-PD(L)1 relapsed/refractory advanced NSCLC HCC	IL-27 possesses direct antitumor activity and indirectly suppresses tumor development by influencing the tumor microenvironment	NCT04374877 NCT05359861
CCL2, CCL3, CCL5	CCR2/CCR5 inhibitor + nivolumab or chemotherapy	Advanced Solid Tumors	Suppression of tumor-associated myeloid cell	NCT03184870
VEGF	Bevacizumab + atezolizumab	untreated metastatic renal cell carcinoma	Suppression of tumor-associated myeloid cell	NCT01984242

Legend: PTCL, peripheral T-cell lymphoma; CTCL, cutaneous T-cell lymphoma; ccRCC, clear cell renal cell carcinoma; HCC, hepatocellular carcinoma; IL-6, interleukin-6; LIF, leukemia inhibitory factor; NSCLC, non-small cell lung cancer; PD(L)1, programmed cell death ligand.

## Data Availability

Not applicable.

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
