# Peer review of "Navigating the Cytokine Seas: Targeting Cytokine Signaling Pathways in Cancer Therapy"

_ijms, 2024, doi:10.3390/ijms25021009_

Round 1
Reviewer 1 Report
Comments and Suggestions for Authors
Overall, this manuscript is written well and informative for readers in this issue, but the context is relatively broad. Therefore, I feel that the authors need to reorganize the manuscript to more clearly focus on both cytokines and cytokine signaling molecules or either ones.
1. The authors stated in the title and abstract that this review article aims to provide an overview of the significance of cytokines in cancer development and therapy by highlighting the importance of targeting cytokine signaling pathways as a potential therapeutic approach. However, the authors mainly described cytokine signaling pathways in the text, but the Table 1 is related to clinical trails on cytokine-based interventions using cytokines or antibodies against cytokines, but not cytokine signaling molecules. If the authors really want to focus on targeting cytokine signaling pathways, Table 1 should be related to clinical trials on cytokine signaling molecules, such as maybe antagonists or agonists to cytokines, as well as cytokine receptors, JAK/STAT, SOCS, NF-kB, Smad, MAPK, Roh/ROCK, PI3K and so on. Therefore, I feel that the authors need to reorganize the manuscript to focus on both cytokines and cytokine signaling molecules or either ones.
2. There is no explanation on the Figure 1 and Table 1 in the text and legends. Therefore, the authors are required to describe more about the contents of Figure 1 and Table 1 in the text by citing individual pathways, molecules, and clinical trials shown in Figure 1 and Table 1.
3. There are many paragraphs even composed of one or two sentences. The authors had better combine some of them into one paragraph if they are related. Or, the authors might want to add subheadings.
4. Some of characters are not converted to symbols. Please confirm them and some typos as well.
Author Response
Overall, this manuscript is written well and informative for readers in this issue, but the context is relatively broad. Therefore, I feel that the authors need to reorganize the manuscript to more clearly focus on both cytokines and cytokine signaling molecules or either ones.
- We appreciate your meticulous review of our manuscript, "Navigating the Cytokine Seas: Targeting Cytokine Signaling Pathways in Cancer Therapy." Your thoughtful comments have been invaluable in enhancing the overall quality and focus of our review. Below are our responses to each of your suggestions.
- We have carefully considered your suggestion to reorganize the manuscript to more clearly focus on either cytokines or cytokine signaling molecules. While we acknowledge the broad context, we would like to emphasize that our intention was to provide a comprehensive overview encompassing targeting cytokines and their receptors in cancer treatment. Cytokines as signaling molecules initiate an intricate intracellular signaling network crucial in neoplastic cell transformation, cancer development and therapy. Our aim was to elucidate how targeting cytokine could be a pivotal therapeutic approach in cancer treatment. We firmly believe that discussing both cytokines and signaling molecules of cytokine induced intracellular signal transduction pathway is essential to present a holistic understanding of the complex cytokine landscape in cancer biology. These elements are interdependent, and alterations in cytokine signal transduction pathways directly impact the role and function of cytokines in cancer progression. However, we provided further citations on small inhibitors of intracellular signal transduction main enzymes to offer readers a thorough comprehension of the multifaceted roles played by cytokines in cancer, from initiation to therapeutic interventions. Once again, we appreciate your thoughtful insights, and we believe that the revisions will further strengthen the manuscript's focus while retaining its comprehensive scope.
- The authors stated in the title and abstract that this review article aims to provide an overview of the significance of cytokines in cancer development and therapy by highlighting the importance of targeting cytokine signaling pathways as a potential therapeutic approach. However, the authors mainly described cytokine signaling pathways in the text, but the Table 1 is related to clinical trails on cytokine-based interventions using cytokines or antibodies against cytokines, but not cytokine signaling molecules. If the authors really want to focus on targeting cytokine signaling pathways, Table 1 should be related to clinical trials on cytokine signaling molecules, such as maybe antagonists or agonists to cytokines, as well as cytokine receptors, JAK/STAT, SOCS, NF-kB, Smad, MAPK, Roh/ROCK, PI3K and so on. Therefore, I feel that the authors need to reorganize the manuscript to focus on both cytokines and cytokine signaling molecules or either ones.
- We acknowledge your insightful feedback on the need to clarify the manuscript's focus on both cytokines and cytokine induced signaling pathway. Our focus was cytokines and their receptors as target for cancer treatment, while signal transduction was described for thorough presentation of the cytokine mechanisms of action. Additionally, there are many papers covering the small molecules for cancer treatment, and we cited the most relevant and significant of them in our paper (i.e., 59. Gu, Y., Mohammad, I. S., Liu, Z. "Overview of the STAT‑3 signaling pathway in cancer and the development of specific inhibitors (Review)". Oncology Letters 19.4 (2020): 2585-2594).
- There is no explanation on the Figure 1 and Table 1 in the text and legends. Therefore, the authors are required to describe more about the contents of Figure 1 and Table 1 in the text by citing individual pathways, molecules, and clinical trials shown in Figure 1 and Table 1.
- Thank you for highlighting the need for a more detailed explanation of Figure 1. In the revised manuscript, we have included a comprehensive description in the text and legends, citing individual pathways, molecules, and clinical trials depicted in these figures. This modification aims to enhance the clarity and relevance of the visual representations.
- There are many paragraphs even composed of one or two sentences. The authors had better combine some of them into one paragraph if they are related. Or, the authors might want to add subheadings.
- We appreciate your suggestion to improve paragraph structure and consider subheadings. Following your advice, we have revise text by simplifying the long sentences to improve coherence.
- Some of characters are not converted to symbols. Please confirm them and some typos as well.
- Your keen observation on character conversion and typos has been duly noted. We have thoroughly reviewed and corrected these issues to ensure the accuracy and professionalism of the manuscript.
- Overall, we believe these revisions have significantly strengthened the manuscript's focus, clarity, and readability. We are confident that the updated version is better aligned with the aims and scope of the journal. Once again, thank you for your time and constructive feedback.
Reviewer 2 Report
Comments and Suggestions for Authors
The manuscript deals with the significance of cytokines in cancer development/therapy and the importance of targeting cytokine signaling pathway as a potential therapeutic approach.
Generally, the manusript is well-written and it emphasizes the most important aspects of signaling pathways involved in cancerogenesis. Besides, the authors provided abundant data and details about interaction mechanisms and cytokines role in cancer promotion/supression.
The very useful part of the manusript is the Part 7. which provide useful data on recent advancements cytokine-based cancer treatment.
There are certain items which need to be added or checked:
-line 98: DAMPS, Please add what is abbreviation for exactly.
-line 163: It seems that the reference Lust et al. 2013 is missing in the Reference list, meaning that reference numbers need to be corrected
line 481: PD(L)1, it is written ''programmed cell''. I suppose you meant Programmed cell death ligand.
-
Author Response
The manuscript deals with the significance of cytokines in cancer development/therapy and the importance of targeting cytokine signaling pathway as a potential therapeutic approach.
Generally, the manusript is well-written and it emphasizes the most important aspects of signaling pathways involved in cancerogenesis. Besides, the authors provided abundant data and details about interaction mechanisms and cytokines role in cancer promotion/supression.
- Thank you for your time and thoughtful review. We sincerely appreciate your positive feedback and constructive comments on our manuscript, "Navigating the Cytokine Seas: Targeting Cytokine Signaling Pathways in Cancer Therapy." Your insights have been instrumental in refining the manuscript further. Here are our responses to your specific suggestions.
The very useful part of the manusript is the Part 7. which provide useful data on recent advancements cytokine-based cancer treatment.
- We are delighted that you found Part 7 of the manuscript particularly useful, and we believe these revisions have further strengthened the manuscript's accuracy and readability.
There are certain items which need to be added or checked:
-line 98: DAMPS, Please add what is abbreviation for exactly.
- Thank you for pointing out the need for clarification. We have now included the abbreviation for DAMPS (Damage-Associated Molecular Patterns) at line 98 to enhance the reader's understanding.
-line 163: It seems that the reference Lust et al. 2013 is missing in the Reference list, meaning that reference numbers need to be corrected
- Your observation regarding the missing reference for Lust et al. (2013) has been duly noted. We have corrected the reference numbers and ensured that Lust et al. (2013) is appropriately included in the Reference list for accurate citation.
line 481: PD(L)1, it is written ''programmed cell''. I suppose you meant Programmed cell death ligand.
- We appreciate your observation on the misspelled terminology used for PD(L)1. The manuscript has been updated to explicitly state "Programmed cell death ligand".
Round 2
Reviewer 1 Report
Comments and Suggestions for Authors
Because the authors have sufficiently addressed to my concerns, I feel now that this manuscript would be suitable for publication.